# Efficacy of Heterologous Vaccination Using Virus-Like Particles and Vaccinia Virus Containing MIC8 and AMA1 Proteins of *Toxoplasma gondii*

**DOI:** 10.3390/vaccines13080862

**Published:** 2025-08-15

**Authors:** Hae-Ji Kang, Fu-Shi Quan

**Affiliations:** 1Department of Microbiology, College of Medicine, Dongguk University, Gyeongju 38066, Republic of Korea; 2Department of Medical Zoology, College of Medicine, Kyung Hee University, Seoul 02447, Republic of Korea; 3Medical Research Center for Bioreaction to Reactive Oxygen Species and Biomedical Science Institute, Core Research Institute (CRI), Kyung Hee University, Seoul 02447, Republic of Korea

**Keywords:** *T. gondii*, MIC8, AMA1, VLPs, rVV, vaccine

## Abstract

**Background:** *Toxoplasma gondii* (*T. gondii*) infection causes serious diseases in immunocompromised patients and causes congenital toxoplasmosis in infants. *T. gondii* microneme protein 8 (MIC8) and apical membrane antigen 1 (AMA1) are essential proteins involved in parasitic invasion. **Methods:** In this study, we generated virus-like particles (VLPs) and recombinant vaccinia virus (rVV) containing MIC8 or AMA1 proteins. Vaccine efficacy was evaluated in mice (BALB/c) upon challenge infection with *T. gondii* ME49. **Results:** Intramuscular immunization with heterologous vaccines (rVV + VLPs; rVV for prime and VLPs for boost) elicited *T. gondii*-specific IgG antibody responses in mice. Four weeks after the boost, all mice were orally challenged with *T. gondii* ME49, and protective immunity was assessed. The responses of antibody-secreting cells for IgG2a and IgG2b and those of memory B cells and CD4+ and CD8+ T cells were higher in the rVV + VLP group than in the VLP + VLP group. The rVV + VLP group exhibited a significant reduction in cyst count in the brain. **Conclusions:** These findings indicate that heterologous vaccination with vaccinia viruses and VLPs improves vaccine efficacy.

## 1. Introduction

*T. gondii* is an obligatory intracellular protozoan parasite with a complex life cycle, infecting approximately one-third of the global population. *T. gondii* establishes chronic infection by differentiating into bradyzoites, which persist within tissue cysts, most notably in the brain and muscles [1,2]. Although *T. gondii* infection is often asymptomatic in immunocompetent individuals, toxoplasmosis can cause life-threatening complications in immunocompromised patients and severe congenital abnormalities when acquired during pregnancy. Given the widespread prevalence and potential severity, an effective vaccine must induce humoral and cellular immune responses capable of targeting the latent stage of the *T. gondii* and suppressing tissue cyst formation to prevent chronic toxoplasmosis. [3,4]. However, vaccine development is further complicated by the *T. gondii* ability to evade host innate immunity through mechanisms such as modulating host cell signaling and forming parasitophorous vacuoles [5,6].

Developing such a vaccine remains challenging, as it requires optimal antigen selection and an appropriate delivery platform. Nevertheless, extensive research is ongoing to identify effective antigen candidates and improve vaccine strategies against *T. gondii*. To address this challenge, MIC8 and AMA1 were selected as antigenic targets in this study based on their critical roles during the invasion process of *T. gondii*. MIC8, a member of the micronemal protein family, is essential for parasite motility and adhesion to host cells, functioning as a transmembrane escorter that directs other adhesins to the parasite surface [7]. AMA1, on the other hand, is secreted from apical organelles and interacts with rhoptry neck proteins to form the moving junction, a specialized structure that facilitates active invasion of host cells [8].

These antigens have shown partial protection in mouse models when delivered via virus-like particle (VLP) platforms, which are valued for their safety and ability to stimulate strong humoral responses without the risk of genome integration or replication [9,10]. However, VLPs are less efficient at delivering antigens into the cytosolic MHC-I pathway, and consequently, their ability to prime robust CD8^+^ T-cell and memory B-cell responses central correlates of resistance to intracellular parasites is limited. Recently, recombinant vaccinia virus (rVV) has emerged as a promising viral vector platform capable of overcoming these limitations [11]. rVVs replicate in the cytoplasm of infected cells, ensuring high-level, endogenous antigen synthesis and presentation via both MHC-I and MHC-II pathways. This feature enables rVVs to induce a balanced, multifunctional response, encompassing strong CD4^+^ and CD8^+^ T-cell activation and durable memory B-cell formation [12,13].

Heterologous prime–boost strategies offer a rational means of combining the complementary immunological advantages of different vaccine platforms. By leveraging the unique strengths of each platform, such as the strong cellular immunity induced by viral vectors and the robust humoral responses elicited by VLP-based vaccines, this approach can amplify both the magnitude and quality of the immune response. In other infectious disease models, including malaria [14], influenza virus [15], and SARS-CoV-2 [16,17], such heterologous regimens have consistently outperformed homologous schedules by enhancing response breadth, magnitude, and durability while often reducing reactogenicity and the required vaccine dose.

Despite the potential benefits, the heterologous vaccination regimen has been underexplored in the context of *T. gondii* ME49 infection. Particularly, few studies have examined whether delivering the same antigens through distinct platforms can synergistically modulate immune responses and improve protection against *T. gondii* ME49 infection. Accordingly, the present study aimed to evaluate the immunogenicity and protective efficacy of a heterologous vaccination regimen using MIC8 and AMA1 on rVV and VLP platforms. This approach was directly compared to a homologous VLP + VLP regimen to determine whether platform diversity enhances the immune response and confers superior protection against *T. gondii* ME49 infection.

## 2. Materials and Methods

### 2.1. Ethical Statement

Six-week-old female BALB/c mice (NARA Biotech, Seoul, Republic of Korea) were maintained under standardized laboratory conditions with unrestricted access to food and water. Prior to the start of this experiment, all animals underwent a four-day acclimatization period. All animal experiments were approved by the Institutional Animal Care and Use Committee of Kyung Hee University (approval no. KHSASP-20–648).

### 2.2. Parasite

*T. gondii* ME49 were maintained in BALB/c mice by long-term serial intraperitoneal passaging. For experimental infections, brain tissues were collected 4 weeks after infection, when cysts are most abundant and morphologically distinct. The brains were homogenized in phosphate-buffered saline (PBS) using a 21-gauge needle to efficiently release cysts with minimal damage. To improve cyst visibility and facilitate isolation from brain debris, the homogenates were further purified using a sucrose gradient, allowing for clearer identification and collection for downstream challenge infection study use [18].

### 2.3. Generation of VLPs and rVVs Presenting T. gondii MIC8 or AMA1

To generate VLPs, sequences of the *T. gondii* antigens MIC8 (GenBank: GQ260152.1) and AMA1 (GenBank: AF010264.1) were cloned into the pFastBac vector and transformed into DH5α and DH10Bac *Escherichia coli* to produce recombinant bacmids. The bacmids were transfected into Sf9 insect cells using Cellfectin™ II Reagent. Five days post-infection, cell enlargement was observed, indicating successful infection and replication of recombinant baculovirus (rBV). rBVs were harvested from the culture supernatant, amplified, and co-infected into Sf9 cells with rBVs expressing the influenza matrix protein 1 (M1) to produce VLPs. The infected Sf9 cells were incubated at 27 °C, and culture supernatants were collected three days later for VLP purification [18]. To generate rVVs, the MIC8 or AMA1 genes were cloned into the vaccinia virus transfer vector pRB21. The resulting constructs were transfected into CV-1 cells with wild-type vaccinia virus. rVVs were selected using a plaque assay in Vero cells [19], and at least five plaques were isolated and amplified. The expression of MIC8 and AMA1 proteins was confirmed by Western blotting and enzyme-linked immunosorbent assay (ELISA), and the morphology of the rVVs and VLPs was verified by transmission electron microscopy (TEM) [20].

### 2.4. Western Blot Analysis

Protein expressions in VLPs and rVVs were visualized and confirmed by Western blotting. VLPs or rVVs were boiled at 95 °C for 5 min, separated on 10% SDS-PAGE gels, and transferred onto PVDF membranes. The membranes were blocked with 5% skim milk in TBST and incubated overnight at 4 °C with a primary antibody derived from sera of mice repeatedly infected (≥3 times) with *T. gondii* ME49 (1:500), which detects MIC8 and AMA1. After washing with TBST, the membranes were incubated with horseradish peroxidase (HRP)-conjugated goat anti-mouse secondary IgG antibody (1:2000) for 2 h, and protein bands were detected using enhanced chemiluminescence [18].

### 2.5. Vaccination and Challenge Infection

The mice were randomly divided into experimental groups (n = 6 per group) and immunized via the intramuscular route at 4-week intervals. For homologous vaccination, mice received two doses of MIC8 VLPs and AMA1 VLPs, which were separately produced and mixed prior to vaccination (total of 80 μg/mouse; 40 μg of each). For heterologous vaccination, mice were primed with a mixture of MIC8 rVVs and AMA1 rVVs (1 × 10^6^ PFU/mouse; 5 × 10^5^ PFU each), followed by a VLP boost using the same formulation as in the homologous group. Blood samples were collected from the retro-orbital sinus 1 and 3 weeks after both the prime and boost immunizations to obtain sera for analysis of *T. gondii*-specific antibody responses. Four weeks after the booster vaccination, all mice were orally challenged with 450 cysts of *T. gondii* ME49 in PBS. The mice were monitored daily for survival and body weight changes for 35 days.

### 2.6. Measurement of T. gondii-Specific IgG in Serum

*T. gondii*-specific IgG responses were measured by ELISA at three time points: after prime, boost, and challenge infections. Ninety-six-well plates were coated with *T. gondii* lysate antigen (4 µg/well) and incubated overnight at 4 °C. After blocking with 0.2% gelatin, serially diluted serum samples (1:150, 1:450, and 1:1350) were added and incubated at 37 °C for 1 h. HRP-conjugated goat anti-mouse IgG (1:2000) was then added, and absorbance was read at 450 nm [18]. To ensure reproducibility, each sample was assayed in duplicate.

### 2.7. Quantification of Antibodies from Antibody-Secreting Cells (ASCs)

To evaluate antibody-secreting splenic cells, splenocytes were isolated from mice 4 weeks after the challenge infection. Ninety-six-well plates were pre-coated with *T. gondii* lysate antigen (4 µg/well) and incubated overnight at 4 °C, followed by blocking with 0.2% gelatin. Splenocytes were plated at a density of 3 × 10^6^ cells/well in 200 µL of RPMI-1640 medium supplemented with 10% fetal bovine serum, 1% penicillin–streptomycin, and incubated for 5 days at 37 °C in a 5% CO_2_ incubator. After incubation, HRP-conjugated secondary antibodies specific for IgG, IgA, IgG1, IgG2a, or IgG2b (diluted 1:1000 in PBST) were added to detect the antibody isotypes secreted by the cultured cells. Plates were read at 450 nm using an ELISA plate reader. The number of antibody-secreting cells (ASCs) was quantified based on absorbance and reported as relative optical density [18].

### 2.8. Flow Cytometry

Four weeks after infection, splenocytes were isolated, passed through a 40 μm cell strainer, and treated with RBC lysis buffer for 10 min at room temperature. After washing, cells were resuspended in FACS buffer (PBS containing 2% FBS and 0.1% sodium azide) and blocked with Fc block (anti-CD16/32, cat#553142) for 30 min at 4 °C in dark. Surface staining was performed using antibodies CD3-FITC, CD4-PE-Cy7, and CD8-PE, and CD19 (PE-Cy7), B220 (FITC), CD27 (PE) and CD62L (APC) (all from BD Biosciences or BioLegend) [21,22]. The cells were fixed in 4% paraformaldehyde and analyzed using a BD Accuri C6 flow cytometer.

### 2.9. Statistical Analysis

All data are presented as a mean standard deviation. Statistical comparisons among groups were performed using one-way analysis of variance (ANOVA), followed by Tukey’s or Bonferroni post-hoc tests, depending on the number and type of comparisons. For experiments involving multiple variables, a two-way ANOVA was applied, followed by an appropriate post-hoc analysis to determine the overall effect of the vaccination regimen. All statistical analyses were performed, and graphs were generated using GraphPad Prism v10.1.2 (GraphPad Software, San Diego, CA, USA). Specific tests and significance thresholds are noted in the corresponding figure legends.

## 3. Results

### 3.1. MIC8 and AMA1 Containing VLPs and rVVs Were Successfully Generated and Characterized

Western blotting was performed to confirm the expression of the target antigens MIC8 and AMA1. As shown in Figure 1A,B, MIC8 (75 kDa) and AMA1 (60 kDa) were successfully expressed at the expected molecular weights in both VLP and rVV samples, demonstrating successful expression independent of the delivery platform. These findings indicated that the MIC8 and AMA1 antigens were effectively expressed on VLPs and rVVs. To further verify the structural formation of the vaccine particles, TEM was conducted. As shown in Figure 1C, VLPs and rVVs resembling influenza virus (spherical structures) and pox viruses (brick-shaped) were observed with diameters ranging from approximately 100 to 200 nm. These results indicated that the particles were successfully assembled and effectively displayed the target antigens on the surface.

### 3.2. Serum IgG Response Was Significantly Higher in the VLP + VLP Group than in the rVV + VLP Group

To evaluate *T. gondii*-specific IgG responses, serum samples were collected according to the immunization and challenge schedule outlined in Figure 2. As shown in Figure 3, the VLP + VLP group exhibited significantly higher IgG responses than the rVV + VLP group following both prime and boost vaccinations. In the VLP + VLP group, IgG levels increased markedly after each dose, indicating a robust booster effect. The rVV + VLP group showed only a slight increase after rVV priming; however, a substantial increase was observed after VLP boost. Both vaccination groups showed statistically significant increases in IgG levels compared to the naïve challenge. At each time, the differences between the VLP + VLP and rVV + VLP groups were statistically significant. These findings indicated that repeated immunization with VLPs was more effective in eliciting *T. gondii*-specific serum IgG responses than the heterologous rVV + VLP regimen.

### 3.3. IgG2a and IgG2b Production by ASCs Was Higher in the rVV + VLP Group than in the VLP + VLP Group

To assess whether vaccination enhanced *T. gondii*-specific antibody secretion by ASCs, splenocytes were cultured in vitro, and antibody levels were measured. As shown in Figure 4A–E, both vaccinated groups (VLP + VLP and rVV + VLP) exhibited significantly elevated levels of total IgG, IgA, IgG2a, and IgG2b compared to both naïve and naïve challenge groups, indicating that vaccination effectively stimulated B cell activation and differentiation into functional ASCs. In contrast, IgG1 levels remained unchanged in all groups. Especially, the rVV + VLP group showed higher levels of Th1-associated isotypes IgG2a and IgG2b than the VLP + VLP group, suggesting that rVV primary vaccination enhanced Th1-type immune responses.

### 3.4. CD4^+^ and CD8^+^ T Cell and Memory B Cell Responses Were Significantly Higher in the rVV + VLP Group

To evaluate the vaccine-induced adaptive immunity, T cell and memory B cell responses were analyzed in the spleen (Figure 5A and Figure 6A). Flow cytometry revealed that CD4^+^ and CD8^+^ T cell frequencies were significantly elevated in both vaccinated groups compared with those in the naïve and naïve challenge. Among the vaccinated groups, the rVV + VLP regimen elicited the highest proportions of both CD4^+^ and CD8^+^ T cells (Figure 5B,C). In parallel, memory B cell responses were also evaluated to assess the humoral memory potential conferred by vaccination. A significant increase in the proportion of memory B cells was observed in both vaccinated groups compared to controls, with the rVV + VLP group exhibiting an appreciably higher frequency than the VLP + VLP group (Figure 6B). These results collectively suggest that the heterologous rVV + VLP regimen is more effective in promoting both cellular and humoral memory responses, which are essential for long-term protection against *T. gondi* ME49 (Figure 6B).

### 3.5. rVV + VLP Vaccination Led to Complete Survival and Reduced Brain Cyst Counts

To assess the protective efficacy of the vaccination regimens against *T. gondii* ME49 infection, survival and brain cysts were evaluated 4 weeks after challenge. As shown in Figure 7A, all mice in the vaccinated groups survived the challenge, demonstrating 100% survival, in contrast to the naïve challenge group, which exhibited complete mortality within the observation period. In addition, both vaccinated groups exhibited a significant reduction in brain cyst counts compared to the naïve challenge group. Strikingly, the rVV + VLP group showed the greatest decrease in brain cyst counts, with an average of only 55 cysts per mouse, compared to 425 in the VLP + VLP group and 1080 in the naïve challenge group. These results indicate that the heterologous rVV + VLP regimen confers superior protective efficacy against *T. gondii* ME49 by more effectively preventing cyst formation following infection.

## 4. Discussion

This study evaluated the immunogenicity and protective efficacy of two vaccination regimens, a homologous VLP + VLP regimen and a heterologous rVV + VLP regimen, both of which express MIC8 and AMA1. Our findings demonstrated that while both regimens elicited significant immune responses and conferred protection, the heterologous rVV + VLP regimen was more effective in enhancing cellular immunity and long-term memory, resulting in superior control of *T. gondii* ME49 infection. These results are consistent with our theoretical expectations that the heterologous regimen could induce humoral and/or cellular immune responses that are comparable to or stronger than those elicited by homologous VLP or rVV regimens.

The results revealed that protein bands for MIC8 and AMA1 seem to run higher in rVVs than the bands in the Western blot for VLPs. This discrepancy is likely due to the differences in post-translational modifications or protein folding between the mammalian cells (rVV) and insect cells (VLP) expression systems. However, since both platforms effectively induced protective immune responses, this variation does not appear to impact the immunogenicity of the antigens.

Our data confirmed that the VLP + VLP group induced significantly higher serum IgG antibody responses against *T. gondii* following vaccination compared to the rVV + VLP group. This finding is consistent with previous reports [18,19] demonstrating the strong humoral immunogenicity of VLPs. VLPs are known to efficiently mimic the repetitive and highly ordered antigenic structure of native viruses, which facilitates potent B cell receptor cross-linking and enhances antigen uptake by antigen-presenting cells [23]. This structural mimicry leads to efficient activation of germinal center reactions, promoting class switching and affinity maturation of antibodies. These intrinsic properties make VLPs highly effective at eliciting strong and sustained humoral immune responses [24], as reflected in the elevated IgG levels observed in the VLP + VLP group in this study. 

To investigate how vaccination influenced the generation of antigen-specific humoral responses at the cellular level, we performed ASC assays using splenocytes. ASC analysis showed that total IgG and IgA responses did not differ significantly between the two groups, and IgG1 levels remained statistically unchanged even when compared to the control group. The absence of enhanced IgA production is likely a consequence of the intramuscular route of immunization [25], which is generally suboptimal for eliciting mucosal immune responses. Moreover, the low levels of IgG1 observed may be attributed to the Th1-biased immune environment induced by rVV priming, which actively suppresses Th2 differentiation and thereby limits the production of Th2-associated isotypes such as IgG1 [26].

Interestingly, however, ASC analysis revealed that the rVV + VLP group exhibited significantly higher levels of IgG2a and IgG2b isotypes compared to the VLP + VLP group. These results further support the notion that the Th1-mediated antibody isotype profile, particularly the elevation of IgG2a and IgG2b, may have been driven by the rVV prime immunization [27]. Th1-mediated immune responses are critically important for the control of intracellular parasites such as *T. gondii* [28,29]. This response is characterized by the production of cytokines like IFN-γ, which activates macrophages and enhances their ability to kill intracellular parasites. In addition, it facilitates effective opsonization and complement activation, further contributing to parasite suppression [30]. This type of Th1-biased immunity, primarily orchestrated by CD4^+^ Th1 and CD8^+^ cytotoxic T cells, is further supported by our flow cytometry data, which showed elevated frequencies of both cell types following a heterologous vaccination regimen

In general, T cell populations in the spleens of *T. gondii*-infected mice are markedly reduced compared to those of uninfected controls [31]. A similar pattern was observed in our study, suggesting that this reduction may result from parasite-induced immunosuppression, including impaired T cell activation, increased apoptosis of activated T cells, and redistribution of T cells to non-lymphoid organs such as the brain, liver, or lungs in response to infection. However, the rVV + VLP immunized group maintained CD4^+^ T cell responses at levels comparable to those of uninfected mice. This preservation is likely due to the strong priming effect of the rVV vector, which, as demonstrated by the ASC results, effectively activates antigen-presenting cells and promotes robust CD4^+^ T cell activation and expansion.

Importantly, CD8^+^ T cell responses in the rVV + VLP group were even higher than those observed in naïve mice, indicating that the heterologous vaccination strategy not only preserved but also enhanced cytotoxic T cell immunity. Such robust CD4^+^ and CD8^+^ T cell responses are critical for host defense against *T. gondii*, as CD4^+^ T cells support B cell activation and germinal center formation, while CD8^+^ T cells directly eliminate infected cells and help control parasite replication. Accordingly, the enhanced T cell responses observed in the heterologous group likely contributed to the improved control of *T. gondii* cyst formation.

These cellular immune responses had a meaningful impact on the activation of memory B cells, which are essential for long-lasting antibody-mediated protection [32]. Flow cytometric analysis using CD19^+^B220^+^CD27^+^CD62L^+^ markers revealed that the rVV + VLP group exhibited a significantly higher frequency of memory B cells compared to the VLP + VLP group. This increase is likely due to stronger CD4^+^ T cell activation induced by rVV priming, which promotes germinal center (GC) formation and supports B cell differentiation [33]. Memory B cells are primarily derived from germinal center B cells during T cell–dependent immune responses. Within the GC, activated B cells undergo clonal expansion, somatic hypermutation, and affinity maturation, after which a subset differentiates into long-lived memory B cells [34]. Although GC B cells were not directly assessed in this study, multiple previous reports have demonstrated that vaccination with *T. gondii* antigens in VLP or rVV formats effectively enhances GC B cell responses [18,19]. These findings support the biological plausibility that the increased memory B cell responses observed in the rVV + VLP group reflect enhanced GC activity promoted by the heterologous immunization strategy. In this context, the findings of this study provide new evidence that the heterologous rVV + VLP regimen not only indirectly promotes germinal center activity but also effectively drives the formation of memory B cells.

In *T. gondii* ME49 infection, the severity of histopathological changes in the brain often correlates with the levels of pro-inflammatory cytokines [35,36]. We have previously observed similar results, consistently demonstrating that the vaccinia virus, virus-like particle, or baculovirus vaccines significantly reduced pro-inflammatory cytokine production in the brain [19,37,38]. Thus, we hypothesize that these vaccines could reduce histopathological changes in the brain. Vaccine research for *T. gondii* faces limitations. The life cycle of *T. gondii* involves multiple hosts and stages, expressing different antigens. A vaccine that can target all antigens across multiple hosts is a significant challenge.

To deepen our understanding and confirm the broader applicability of these results, further research is necessary. First, the durability of vaccine-induced immunity over extended periods should be assessed through long-term follow-up studies to determine the persistence of protection and immunological memory. Second, intracellular cytokine staining would provide deeper insights into the functional quality of CD4^+^ and CD8^+^ T cell responses, particularly in terms of cytokine production and effector function. Finally, it will be important to evaluate the cross-protective efficacy of the vaccine regimen against other *T. gondii* strains with varying levels of virulence, particularly those associated with acute infection. Given that MIC8 and AMA1 are invasion-related antigens, further studies should determine whether the heterologous vaccination strategy provides broad protection against strains with differing invasion capabilities.

## 5. Conclusions

Collectively, our findings indicate that the heterologous vaccination regimen combining rVV and VLPs containing *T. gondii* MIC8 and AMA1 confers better protective efficacy compared with a homologous VLP + VLP regimen. By leveraging the complementary strengths of two distinct vaccine platforms, this approach broadens and amplifies both humoral and cellular immunity, thereby establishing a solid foundation for a next-generation vaccination approach.

## Figures and Tables

**Figure 1 vaccines-13-00862-f001:**
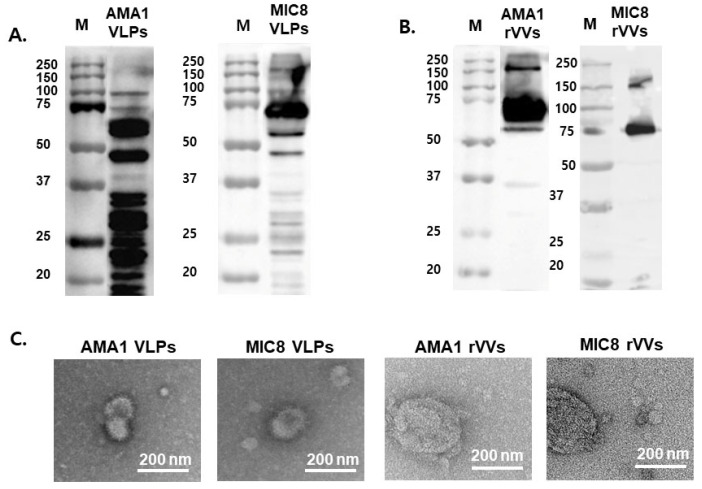
Characterization of virus-like particles (VLPs) and recombinant vaccinia viruses (rVVs). The expression of microneme protein 8 (MIC8) (75 kDa) and apical membrane antigen 1 (AMA1) (60 kDa) proteins in VLPs (**A**) and rVVs (**B**) was confirmed by Western blotting using mouse anti-sera against *T. gondii*. (**C**) Morphological characterization of AMA1 VLPs, MIC8 VLPs, AMA1 rVVs, and MIC8 rVVs by TEM. M: molecular weight marker. Scale bars: 200 nm.

**Figure 2 vaccines-13-00862-f002:**
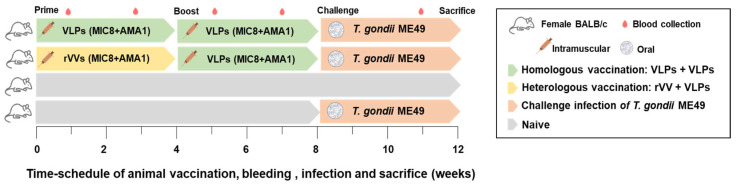
Vaccination schedule and experimental design. Female BALB/c mice (n = 6 per group) were immunized intramuscularly with either VLP + VLP (green + green) or rVV + VLP (yellow + green) and orally challenged with *T. gondii* ME49 (orange). Blood samples (red droplet icon) were collected at designated time points, and mice were sacrificed for analysis 4 weeks after the challenge infection.

**Figure 3 vaccines-13-00862-f003:**
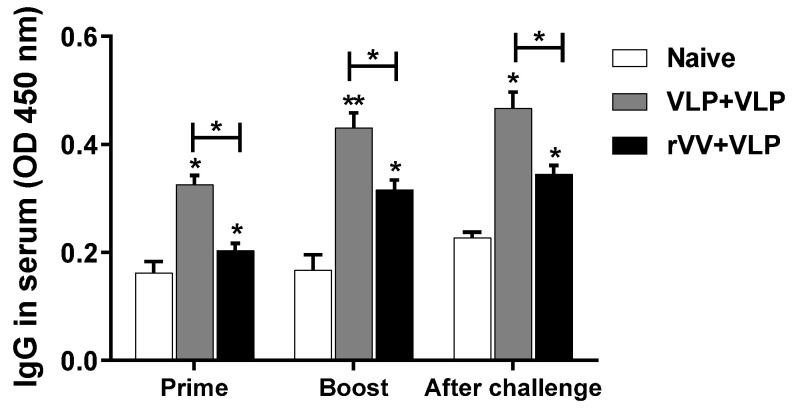
Serum IgG responses after vaccination. *T. gondii*-specific total IgG levels in mouse sera were assessed by ELISA at three time points: after the prime, after the boost, and after oral challenge with *T. gondii* ME49. Data represent mean ± SD. Statistical significance: * *p* < 0.05 and ** *p* < 0.01.

**Figure 4 vaccines-13-00862-f004:**
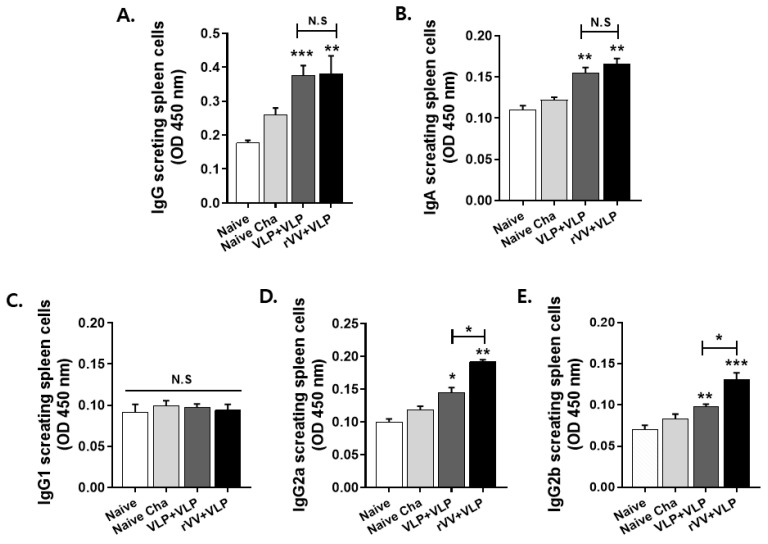
*T. gondii*-specific antibody secretion by antibody-secreting cells. Splenocytes were harvested 4 weeks after challenge and cultured for 5 days. Secreted antibodies were measured using an ELISA assay. Levels of total IgG (**A**), IgA (**B**), IgG1 (**C**), IgG2a (**D**), and IgG2b (**E**) are shown. Data represent mean ± standard deviation. Statistical significance: * *p* < 0.05, ** *p* < 0.01, and *** *p* < 0.001; ns, not significant.

**Figure 5 vaccines-13-00862-f005:**
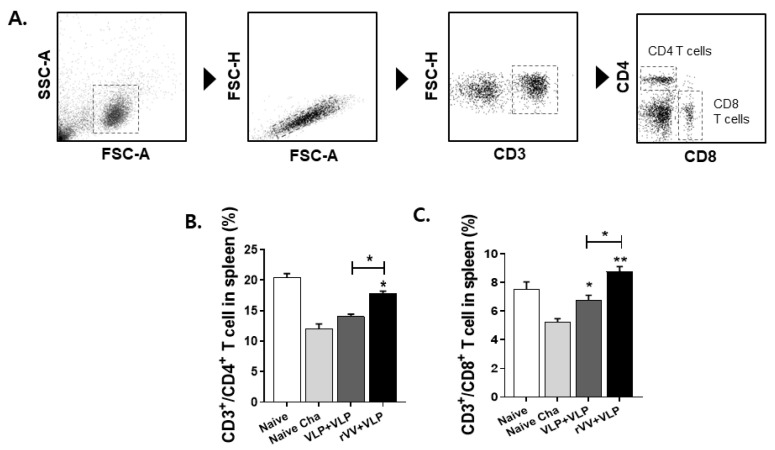
Induction of CD4^+^ and CD8^+^ T cells. Splenocytes were gated (**A**). The frequencies of CD4^+^ T cells (**B**) and CD8^+^ T cells (**C**) among CD3^+^ splenocytes were analyzed by flow cytometry using a standard gating strategy (CD3^+^CD4^+^ and CD3^+^CD8^+^). Data represent mean ± SD. Statistical significance: * *p* < 0.05 and ** *p* < 0.01.

**Figure 6 vaccines-13-00862-f006:**
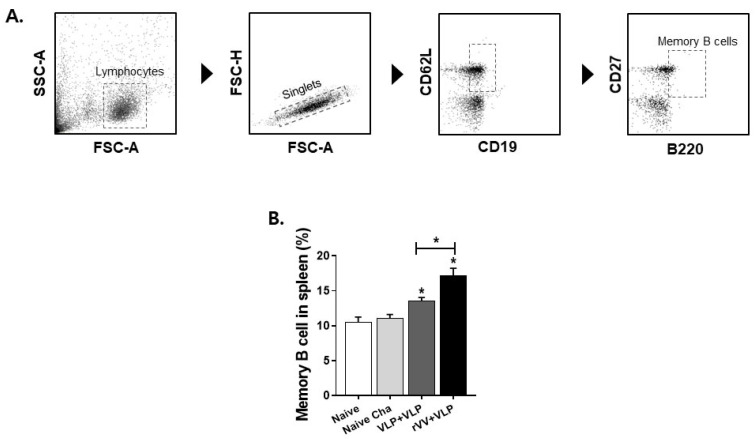
Induction of memory B cells. Memory B cells in the spleen were analyzed by flow cytometry using a gating strategy using CD19^+^, B220^+^, CD27^+^, and CD62L^+^ cells (**A**), and the quantified proportions of memory B cells (**B**). Data represent mean ± SD. Statistical significance: * *p* < 0.05.

**Figure 7 vaccines-13-00862-f007:**
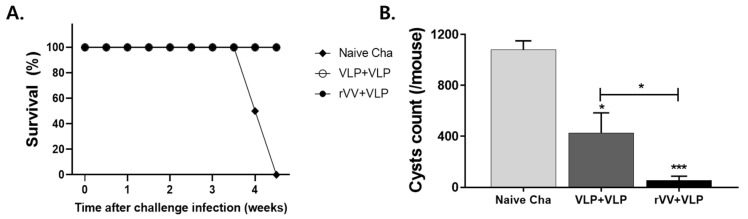
Protective efficacy against *T. gondii* ME49 infection. Survival (**A**) and brain cyst counts (**B**) were confirmed 4 weeks after challenge with *T. gondii* ME49 to evaluate the protective efficacy of each vaccination regimen. Data represent mean ± SD. Statistical significance: * *p* < 0.05 and *** *p* < 0.001.

## Data Availability

Data will be made available on request.

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
