# Peer review of "Efficacy of Heterologous Vaccination Using Virus-Like Particles and Vaccinia Virus Containing MIC8 and AMA1 Proteins of Toxoplasma gondii"

_vaccines, 2025, doi:10.3390/vaccines13080862_

Round 1
Reviewer 1 Report
Comments and Suggestions for Authors
The authors propose a project involving vaccination against Toxoplasma gondii using virus-like strains expressing recombinant MIC8 and AMA1 proteins. The results obtained are quite interesting, but some points need clarification and even improvement, in my opinion:
1) The authors used BALB/c mice in all procedures, but it is known that this strain is genetically more resistant to T. gondii cyst formation. Wouldn't this compromise the assessment of the challenge quality?
2) Are the same results obtained using, for example, the C57Bl/6j strain?
3) It would be very interesting to complement the article with histopathological images of brain tissue after challenge, for visual assessment of cysts and local tissue swelling compared to unvaccinated mice.
Author Response
The authors propose a project involving vaccination against Toxoplasma gondii using virus-like strains expressing recombinant MIC8 and AMA1 proteins. The results obtained are quite interesting, but some points need clarification and even improvement, in my opinion:
- The authors used BALB/c mice in all procedures, but it is known that this strain is genetically more resistant to T. gondii cyst formation. Wouldn't this compromise the assessment of the challenge quality?
Response: Although BALB/c mice are known to exhibit relatively higher resistance to T. gondii cyst formation compared to strains such as C57BL/6, they are widely used in vaccine studies due to their well-characterized immune responses and consistent susceptibility to infection. In our study, we used an established oral challenge dose of 450 ME49 cysts to ensure sufficient infection for evaluating vaccine efficacy. This approach is supported by multiple previous studies, which validate the suitability of BALB/c mice for assessing protective immunity against T. gondii (PLoS ONE 18(4): e0283928; Pharmaceutics 2020, 12, 989; PLoS ONE 14 (8): e0220865; https://doi.org/10.1016/j.micpath.2020.104090).
- Are the same results obtained using, for example, the C57Bl/6j strain?
Response: While we did not evaluate the vaccine in C57BL/6J mice, studies by other researchers have shown that this strain is more susceptible to T. gondii infection compared to BALB/c mice, and lower challenge doses are often used to achieve similar levels of pathology or mortality. Therefore, we expect comparable protective trends; however, further studies are needed to directly assess efficacy in this strain and confirm the generalizability of our findings (Acta Tropica Volume 196, August 2019, Pages 1-6, Parasites & Vectors volume 12, Article number: 140 (2019).
3) It would be very interesting to complement the article with histopathological images of brain tissue after challenge, for visual assessment of cysts and local tissue swelling compared to unvaccinated mice.
Response: We agree that histopathological images would provide valuable visual evidence of cyst burden and tissue inflammation. However, in this study, we focused primarily on quantitative cyst enumeration, which is a widely accepted method for evaluating vaccine efficacy against T. gondii, and thus did not include histopathological analysis. We plan to incorporate such assessments in future studies to offer a more comprehensive evaluation of tissue pathology.
Reviewer 2 Report
Comments and Suggestions for Authors
The comments are in the text of manuscript.

Author Response
Responses: We have revised the manuscript as commented (lines 51, 53, 119).
For the comment in line 126, we previously reported that 5 x 103 PFU of dose was administered by intranasal immunization (https://doi.org/10.1016/j.exppara.2025.108900), which could elicit mucosal immunity, contributing to the vaccine efficacy. In the current study, vaccine was administered by intramuscular immunization (IM) which wouldn’t induce mucosal immunity. Thus, to elicit higher immune response, higher dose (1 x 106) of rVVs was used.
Reviewer 3 Report
Comments and Suggestions for Authors
The manuscript, “Vaccine efficacy induced by heterologous vaccination with virus-like particle and vaccinia virus expressing Toxoplasma gondii MIC8 and AMA1” by Kang et al., describes strategy for developping an effective heterologus vaccine against T. gondii. The authors produced VLPs and VV having MIC8 and AMA1 proteins of T. gondii and immunized BALB mice to test their efficacy. The specific antibody secretion (IgG2a and IgG2b) and reduction in cyst formation confirmed better efficacy in rVV + VLP group. The manuscript is well written and presents all the results clearly. The authors are suggested to address the following concerns:
- The title needs more clarity. I suggest the following: “Efficasy of heterologous vaccination using virus-like particles and vaccinia virus containing MIC8 and AMA1 proteins of Toxoplasma gondii”. The authors may modify it further. Please avoid using term “expressing” as VLP or virus do not express but package MIC8 and AMA1 proteins.
- Protein bands for MIC8 and AMA1 in seem to run higher in rVVs than the bands in western blot for VLPs. Authors must comment these discrepancies and in the discussion.
- Please include the TEM images for VLP and VV without any gondii proteins (as reference images for VLP and VV structure).
Author Response
The manuscript, “Vaccine efficacy induced by heterologous vaccination with virus-like particle and vaccinia virus expressing Toxoplasma gondii MIC8 and AMA1” by Kang et al., describes strategy for developping an effective heterologus vaccine against T. gondii. The authors produced VLPs and VV having MIC8 and AMA1 proteins of T. gondii and immunized BALB mice to test their efficacy. The specific antibody secretion (IgG2a and IgG2b) and reduction in cyst formation confirmed better efficacy in rVV + VLP group. The manuscript is well written and presents all the results clearly. The authors are suggested to address the following concerns:
Response: We have revised the title and manuscript as commented (lines 2, 3, 17, 169).
- Protein bands for MIC8 and AMA1 in seem to run higher in rVVs than the bands in western blot for VLPs. Authors must comment these discrepancies and in the discussion.
Response: We have newly added comment on discrepancies of western bands between rVVs and VLPs in the discussion section (lines 286–289). This discrepancy is likely due to the differences in post-translational modifications or protein folding between the mammalian cells (rVV) and insect cells (VLP) expression systems.
Please include the TEM images for VLP and VV without any gondii proteins (as reference images for VLP and VV structure).
Response: In our previous study, we have presented TEM images for M1VLPs without vaccine antigen protein (https://doi.org/10.3390/pathogens10101291, Fig 1E). In this study, we did not obtain TEM images of VLPs or rVVs without T. gondii antigens. However, based on previous studies (Curr Protoc. 2021 Mar; 1(3):e55. doi: 10.1002/cpz1.55), VLPs that do not express T. gondii antigens typically exhibit a smooth, spherical morphology consistent with the M1-based structural scaffold. In the case of rVVs, the presence or absence of antigen inserts does not noticeably alter their overall morphology, and thus structural differences are generally indistinguishable by TEM.
Round 2
Reviewer 1 Report
Comments and Suggestions for Authors
Evaluating the authors' responses, I understand their rationale regarding the mouse models used and understand that, since this study was not tested in C57 mice, it is difficult to add this data at this time.
Regarding the histopathology data, however, I believe it is essential to complement the authors' findings. Pathological evaluation is crucial to confirm their findings.
Comments on the Quality of English LanguageI don't see any problems with writing.
Author Response
Reviewer 1
Evaluating the authors' responses, I understand their rationale regarding the mouse models used and understand that, since this study was not tested in C57 mice, it is difficult to add this data at this time.
Regarding the histopathology data, however, I believe it is essential to complement the authors' findings. Pathological evaluation is crucial to confirm their findings.
Response: Thank you for the comment.
In T. gondii infection, especially in the brain (as seen in toxoplasmic encephalitis or during chronic latent infection), the severity of histopathological changes often correlates with the levels of pro-inflammatory cytokines [Eur J Microbiol Immunol (Bp); Parasit Vectors. 2022 Dec 24;15:487].
In our previous study, we reported that a recombinant vaccinia virus targeting T. gondii ROP4 significantly reduced the production of inflammatory cytokines IFN-γ and IL-6 in the brain [https://doi.org/10.3390/vaccines10020152]. Additionally, we demonstrated that recombinant vaccinia virus expressing Plasmodium berghei apical membrane antigen 1 or microneme protein led to a reduction of IFN- γ and IL-6 levels in the spleen [https://doi.org/10.3390/tropicalmed7110350].
In our other T. gondii vaccine studies, we consistently observed similar results, demonstrating the virus-like particles/baculovirus vaccines significantly reduced pro-inflammatory cytokine production. (https://doi.org/10.1080/17435889.2024.2403333; https://doi.org/10.1016/j.actatropica.2024.107501; https://doi.org/10.1371/journal.pone.0283928; https://doi.org/10.3390/vaccines10101588; doi: 10.3389/fcimb.2021.735191).
In summary, the vaccines in our previous studies have effects on reducing inflammatory cytokine production, and based on that, we decided not to repeat the assay in the current study.
Reviewer 3 Report
Comments and Suggestions for Authors
The authors have effectively addressed the reviewer’s concerns in the revised manuscript. Therefore, I do not have any further comment.
Author Response
Reviewer 2
The authors have effectively addressed the reviewer’s concerns in the revised manuscript. Therefore, I do not have any further comment.
Response: Thank you so much!